# Silanized Graphene Oxide-Supported Pd Nanoparticles and Silicone Rubber for Enhanced Hydrogen Elimination

**DOI:** 10.3390/ma15134578

**Published:** 2022-06-29

**Authors:** Yu Wang, Tao Xing, Yongqi Deng, Kefu Zhang, Yihan Wu, Lifeng Yan

**Affiliations:** 1Department of Chemical Physics, University of Science and Technology of China, Hefei 230026, China; wy1998@mail.ustc.edu.cn (Y.W.); ddyyqq@mail.ustc.edu.cn (Y.D.); zkefu@mail.ustc.edu.cn (K.Z.); yihanwu@mail.ustc.edu.cn (Y.W.); 2Institute of System and Engineering, China Academy of Engineering Physics, 64 Mianshan Road, Mianyang 621900, China

**Keywords:** polysiloxane, modified graphene, hydrogen getter, self-healing

## Abstract

Hydrogen is a dangerous gas because it reacts very easily with oxygen to explode, and the accumulation of hydrogen in confined spaces is a safety hazard. Composites consisting of polymers and catalysts are a common getter, where the commonly used catalyst is usually commercial Pd/C. However, it often shows poor compatibility with polymers, making it difficult to form a homogeneous and stable composite. In this work, palladium chloride (PdCl_2_) was converted to palladium (Pd) nanoparticles by reduction reaction and supported on graphene oxide (GO) modified by silanization. Spherical Pd nanoparticles with a size of 2–36 nm were uniformly distributed over the Silanized graphene oxide (SGO) matrix. When mixed with Pd/SGO, polymethylvinylsiloxane can be cured to silicone rubber (SR) by B_2_O_3_. Afterwards, the vinyl in the polymer can interact with hydrogen under the catalysis of Pd through the addition reaction, thus achieving the purpose of hydrogen elimination. The polymer elastomers with excellent self-healing properties and improved hydrogen elimination performance were prepared and were superior to the commercial Pd/C. In addition, excellent environmental adaptability was also demonstrated. The new getter SR-Pd/SGO provides a new avenue for developing polymer getters with superior properties.

## 1. Introduction

Hydrogen is an attractive fuel due to its important properties of energy efficiency and environmental cleanliness [1]. However, hydrogen is a flammable and explosive gas: when the volume fraction of hydrogen in the air reaches 4%, it may explode. Moreover, if hydrogen is not eliminated, it will also cause hazards such as hydrogen corrosion, degradation of electronic components, and embrittlement of materials [2]. In some specific environments, the concentration of hydrogen needs to be maintained below a few percent to ensure safety [3]. Avoiding the accumulation of hydrogen in confined spaces has become a worthy research direction, while most hydrogen storage devices cannot absorb hydrogen independently [1].

One of the most effective types of getters is a mixture of alkynes and Pd/C catalysts. This getter can remove hydrogen through the hydrogenation reaction of alkynes. The hydrogenation of these alkynes is a characteristic of their fast, irreversible, and high-capacity hydrogen uptake rates [4,5]. Elizabeth et al. reported the hydrogen uptake kinetics of 1,4-bis(phenylethynyl)benzene (DEB) mixed with Pd on activated carbon [6]. However, such getters are usually free-flowing powders, which are difficult to apply in a different environment. Therefore, polymer getters were developed to solve this problem. Polymer getters will work better in certain conditions because they can be easily fabricated into various shapes and forms due to their processability [4,7]. A common polymer getter is composed of polydimethylsiloxane (PDMS), Pd/C, and DEB [4]. Compared with powder getters, although this polymer getter has processability, the capacity of the getter is limited to the content of DEB and the incompatibility between the three components is difficult to solve. Therefore, the recorded performance of polymer getters suggests that it is inadequate for use and remains to be improved.

In this work, polymethylvinylsiloxane (PMVS) was adopted as the polymer matrix and simultaneously took on the role of hydrogen absorption. PMVS performed good self-healing properties after curing into silicone rubber (SR) with B_2_O_3_. However, the commercial Pd/C tended to agglomerate in SR, which will reduce the active reaction sites of Pd, making this common catalyst unsuitable for SR. However, Pd supported on the surface of GO is a common catalyst, which can make Pd nanoparticles more uniformly dispersed and increase reaction sites [2]. GO has the advantages of low density, high specific surface area, multiple reactive sites, and good chemical stability [8,9]. GO introduced into SR as a filler can endow the composite with better mechanical, electrical, thermal, and electromagnetic properties [3,4]. However, GO directly added to silicone rubber will also have poor compatibility [5]. The surface of GO contains many carboxyl, hydroxyl, and epoxy groups, which can be used for modification [10,11]. GO was modified into SGO, which can be uniformly dispersed in SR [12]. Then, PdCl_2_ was converted into Pd nanoparticles, which were supported on SGO through the reduction in NaBH_4_ to provide more reaction sites [1]. Figure 1 shows that hydrogen molecules diffused into polymer getters and were adsorbed on Pd nanoparticles. Then the hydrogen molecule was dissociated into two hydrogen atoms under ambient temperature, and an additional reaction further occurred between the vinyl group in SR and the hydrogen atom. This new getter was a homogeneous and stable composite with self-healing properties and high-capacity hydrogen uptake, making it an ideal hydrogen scavenger in various complex environments.

## 2. Materials and Methods

### 2.1. Materials

Dichloromethylvinylsilane, Pd/C, and KI were sourced from Aladdin Biochemical Technology Co., Ltd. (Shanghai, China). B_2_O_3_ was sourced from Energy Chemical Inc. (Shanghai, China). Phenyldimethylchlorosilane was sourced from Bide Medicine Inc. (Shanghai, China). Pd/C was sourced from Macklin Chemical Inc. (Shanghai, China). All other chemicals were sourced from Sinopharm Chemical Reagent Co., Ltd. (Shanghai, China).

### 2.2. Preparation of Methyl Vinyl Silicone Oil

For the preparation of methyl vinyl silicone oil, 20 mL dichloromethylvinylsilane and 20 mL diethyl ether were both added to a 250 mL round bottom flask under an ice-water bath. With stirring, 40 mL of water was slowly added dropwise (the reaction proceeds vigorously if the water is added too quickly) and the mixture was reacted at room temperature for 12 h. The reaction equation is shown in Figure 2. After the hydrolysis polycondensation was completed, a separatory funnel was used to separate the liquid layers. The upper layer was the organic layer, and the lower layer was the water layer. After separating the organic layer, the diethyl ether was rotary evaporated to obtain polymethyl vinyl silicone oil.

### 2.3. Preparation of Pd/SGO

GO was produced by the modified Hummers method [13,14]. First, 250 mg GO was added to a 250 mL round-bottomed flask, and then an appropriate amount of water was added for ultrasonic dispersion to dilute the GO concentration to 2 mg/mL. Then, 2 mL of N-(2-aminoethyl)-3-aminopropyltrimethoxysilane (AEAPTMS) was added and continued sonication for 10 min. At 98 °C, the mixture was stirred and refluxed for 4 h. After the reaction, the mixture was filtered with suction, and then washed three times with water and absolute ethanol. It was then poured into 200 mL of absolute ethanol for ultrasonic dispersion, 7.5 mL of TEOS and 5 mL of water were added, and the mixture was stirred at room temperature for 4 h. Then, ammonia water was added dropwise to adjust the pH to 9–10. The mixture was stirred at 60 °C for 2 h and 40 °C for 4 h. Finally, after washing with water and absolute ethanol until neutral, it was dried at 80 °C to obtain SGO.

Next, 0.1 g of SGO, 250 µL of HCl, and 50 mL of an aqueous solution containing 9 mg of PdCl_2_ were added to a 100 mL flask. After sonicating the solution for 20 min, the solution was stirred at 70 °C for 15 min. Then, 0.15 g of sodium borohydride (NaBH_4_) was slowly added to the vessel and stirred for 20 min. Finally, after centrifuging the solution, the obtained precipitate was washed three times and then dried at a constant temperature of 80 °C to obtain 5% Pd/SGO. SGO with 3%, 8%, 10%, and 12% Pd content was prepared by the same procedure described above by adjusting the PdCl_2_ content in the aqueous solution.

### 2.4. Preparation of the Getter

To prepare the getter, 1 g polymethyl vinyl silicone oil and 0.06 g B_2_O_3_ were added to a 10 mL round bottom flask. Then, 0.1 g 3% Pd/SGO was added to the composite, where the product was numbered G1. After sonicating for 10 min and stirring for five minutes, it was heated to 100 °C for 6h to cure SR. Products G2, G3, G4 and G5 were prepared by the same procedure described above by replacing 0.1 g 3% Pd/SGO with 0.1 g 5% Pd/C (G2), 0.05 g 8% Pd/SGO (G3), 0.05 g 10% Pd/SGO (G4) and 0.05 g 12% Pd/SGO (G5), respectively. The physical images of pure SR and other samples were shown in Appendix A.

### 2.5. Quantification of Vinyl Content

To determine the vinyl content, a titration experiment was used. Next, 40 mg polymethyl vinyl silicone oil was dissolved in 5 mL CCl_4_, which was added into a 10 mL round bottom flask. Then, 0.4 mL of the prepared IBr solution (2 mmol/mL in CCl_4_) was added to the silicone oil solution, which was stirred at room temperature for 120 min. Finally, 4 mL of the prepared IK solution (0.2 mmol/mL in CCl_4_) was added to the vessel and stirred vigorously for 15 min. One drop of aqueous starch solution was added as an indicator and 0.2 mmol/mL NaS_2_O_3_ solution was used to titrate this mixture. The NaS_2_O_3_ solution consumed at the end of the titration (when the color of the mixture changes from blue-violet to colorless) was recorded. Simultaneously, a blank test without the sample was performed, and the consumed NaS_2_O_3_ solution was subtracted from the sample consumption to give the actual consumption of each sample. Titration experiments were performed three times and the average was taken to calculate the vinyl content. The final result is shown in Table 1. There is little difference between the actual content of the vinyl and the theoretical content.

### 2.6. Quantification of Molecular Weight

For quantification of the average molecular weight, the hydroxyl group at the end of the PMVS molecular chain was replaced by a phenyl group. Next, 0.7 g chlorodimethyl(phenyl)silane and 1 g PMVS were added to a 10 mL round bottom flask and stirred for two hours. Figure 3 shows that the phenyl group is grafted into the polymer by the reaction of the chlorine atom with the hydroxyl group. Both the product and chlorodimethyl(phenyl)silane were subjected to ^1^H NMR spectroscopy, as shown in Appendix A. The molecular weight was calculated by comparing the areas of the vinyl and phenyl peaks on the molecular chain, as shown in the following Equation (1):(1)$$M_{1}=\frac{n_{1}S_{2}}{n_{2}S_{1}}\times M_{2},$$
where *M*_1_ is the molecular weight of the polymer, *n*_1_ is the number of hydrogen atoms in the two phenyl groups at the end of each molecular chain of the product, *n*_2_ is the number of hydrogen atoms in the vinyl group, $\frac{S_{2}}{S_{1}}$ is the ratio of the phenyl peak area and the vinyl peak area in the ^1^H NMR spectrum of the product, and *M_2_* is the molecular weight of the monomer.

### 2.7. Absorption Test in Pure H_2_

A homemade apparatus is shown in Appendix A for conducting absorption tests in pure H_2_ atmospheres. By using an L300500i helium mass spectrometer leak detector (Leybold, Koln, Germany), it was determined that the leakage rate of the apparatus was less than 10^−9^ Pa m^3^/s, which was suitable for the test. The sample was placed in a stainless-steel reactor (Xuwei technology Inc., Chengdu, China) with a 100 mL volume, where the reactor was then closed and a vacuum was applied to the apparatus (valves 1 off, and 2, 3, and 4 on). When the internal pressure of the gas reservoir was pumped to 100 Pa, the vacuum pump and the reactor were disconnected (valves 4 and 3 off), and H_2_ from the hydrogen tank was slowly introduced to the device (valves 1 and 2 on) until the internal pressure reached 200 KPa. The reactor was reconnected (valve 1 off and 3 on), and H_2_ was introduced into the reactor. Then, the data acquisition software (SUPY1.0, 2015, Simingte Inc., Hangzhou, China) was used to record and analyze the data. The pressure change was recorded as a function of time and the capacity (volume of hydrogen absorbed per SR or catalyst weight, mL/g) was calculated from the pressure drop according to the ideal gas equation (Equation (2)):(2)$$Capacity=\frac{\left(P_{1}V_{1}-P_{2}\left(V_{1}+V_{2}-V_{3}\right)\right)\times 22400}{mRT},$$
where *m* is the weight of the *SR* or the catalyst, *T* is temperature, *R* is the ideal gas constant, and *V*_1_ is the volume of the gas reservoir and tubing between valve 1 and two containers (gas reservoir and reactor), which was calibrated to be 312.0 mL. *V*_2_ is the volume of the reactor, which was calibrated to be 114.6 mL, and *V*_3_ is the volume of the sample, which was about 0.8 mL when a 1.0 g sample was tested.

## 3. Results and Discussion

### 3.1. Synthesis and Characterization of SGO

The surface of GO was abundant in hydroxyls, carboxyl, and epoxy groups. At high temperatures. Figure 4 shows that the amino groups at one end of AEAPTMS will react with the functional groups on the surface of GO or form non-covalent bonds with GO nanosheets [11], and the three methoxy groups at the other end of AEAPTMS will be hydrolyzed into Si-OH, which can form a monolayer through hydrogen bonding interaction. As a result, GO was initially modified by reduction, and then a large number of Si-OH were exposed on the surface of GO. Figure 5 showed that the four ethoxy groups of TEOS can be hydrolyzed into four Si-OH. After the initial modified GO was dispersed in ethanol, the appropriate amount of water and TEOS were added. Under alkaline conditions, the four Si-OH obtained by the hydrolysis of TEOS condensed with the Si-OH on the surface of the initial modified GO, which made the siloxane on the GO nanosheet surface condense into polysiloxane. Therefore, polysiloxane was covered on the surface of GO, so that GO was uniformly silanized to form SGO.

In order to verify the changes in the related functional groups in GO and SGO, infrared spectra were performed as shown in Figure 6. In the infrared spectrum of GO, there are obvious absorption peaks at 3354 cm^−1^, 1575 cm^−1^, 1408 cm^−1,^ and 1065 cm^−1^, which correspond to the stretching vibration of the O-H bond and C=O bond on the surface of GO, skeleton vibration of C=C and stretching vibration of C-O bond in GO, respectively. After GO was converted into SGO, there existed several new characteristic absorption peaks. There were strong absorption peaks at 1211 cm^−1^ and 1078 cm^−1^, which were attributed to the stretching vibrations of Si-O-C and Si-O-Si bonds. Additionally, a characteristic absorption peak was located in 456 cm^−1^, which was the torsional vibration of the Si-O-Si bond [15,16]. This indicated that the surface of GO has been successfully grafted with polysiloxane.

In order to characterize the vibrations caused by the changes in the geometry and chemical bonds of molecules, Raman spectra were performed [17]. As can be seen from Figure 7, all samples contain two characteristic peaks of graphene, namely the D band at 1345 cm^−1^ and the G band at 1595 cm^−1^ [18,19]. The Raman spectra show no obvious shift at peak position from GO to Pd/SGO. The intensity ratio of D/G bands determines the number of defects in the graphene structure [6]. Compared to that of GO (I_D_/I_G_ = 0.85), the D/G intensity ratio significantly increased in the SGO (I_D_/I_G_ = 1.88), and then slightly increased in the 5% Pd/SGO (I_D_/I_G_ = 1.94), suggesting grafting of polysiloxane onto GO surface can increase the defects in graphene and the support of Pd, probably resulting in the formation of Pd–C and Pd–O–C bonds [20]. In conclusion, Raman spectra indicated that GO was converted into SGO and Pd nanoparticles were supported on the SGO surface.

In order to further explore the chemical composition of GO and SGO, X-ray photoelectron spectroscopy (XPS) analysis was applied to investigate their elemental composition and states [7]. Table 2 shows the C, O, N, and Si atomic ratios for each sample. It can be seen that the content of Si, O, and N in SGO increased and the content of C decreased. Figure 8a shows the XPS wide scan of GO and SGO. Compared with GO, the peaks of Si2s and Si2p have increased in the wide scan spectra of SGO, and the intensity of the O1s peak was increased. Figure 8b,c shows the C1s narrow scan of GO and SGO. Compared with GO, it can be clearly seen that the peak intensity of the C-O bond was significantly reduced, while the peak of the C-N bond (286.3 eV) appeared, which confirmed the epoxy groups on the GO nanosheets can react with the amino group of AEAPTMS. The appearance of four new peaks in Figure 8d further confirmed that GO was converted into SGO.

In order to characterize the microstructures of GO, SGO, and Pd/SGO, transmission electron microscopy (TEM) was performed. In Figure 9a,b, the overall transparency of GO was still high and some folds occurred, which was reduced in SGO due to the modification and silanization of the GO, as shown in Figure 9c,d. Many shadows were shown on the surface of the lamellae, which were uniformly distributed on the lamellae, indicating that the polysiloxane was uniformly grafted on the GO surface. The images of Pd nanoparticles/SGO hybrid at various magnifications in Figure 9e,f indicate that Pd nanoparticles were uniformly distributed on the surface of SGO without agglomeration. Appendix A showed that the particle size of the Pd nanoparticles supported on SGO was mostly about 2–12 nm after the statistics, where the distance of each group was 3.4 nm.

### 3.2. Characterization of SR-Pd/SGO

To compare the hydrogen uptake capacity between SR-Pd/C and SR-Pd/SGO, an absorption test in pure H_2_ was performed. First, 1 g of each sample was placed into the hydrogen absorption test apparatus. The obtained data were used to construct the time-pressure curves, as shown in Figure 10. The final calculation results of the hydrogen absorption test are shown in Table 3, and were calculated by Equation (2). Compared with G3, the hydrogen absorption (/g SR) performance of all samples was obviously improved, except G1 sample, which was caused by the low Pd concentration of G1. Even though the Pd content of G4 is the same as that of G3, the hydrogen absorption (/g SR) capacity was increased by 23.31%. Compared with G3, the hydrogen absorption (/mg catalyst) performance of all the samples was obviously improved. However, with the increase in Pd concentration, the improved capacity (compared to G3) was gradually reduced, which was caused by the catalytic efficiency of a single Pd atom decreasing with the increase in Pd concentration. It can be seen that the catalytic performance of Pd/SGO was greatly improved compared with that of commercial catalysts. Since the G5 sample had the best hydrogen absorption performance, it was adopted as the other subsequent tests.

In order to observe the surface topography of SR-Pd/C and SR-Pd/SGO, the scanning electron microscope (SEM) was performed. In Figure 11a, the Pd/C particles were not uniformly dispersed in the SR cross-section with easy agglomeration, resulting in the inability to form a homogeneous composite. However, the dispersion of Pd/SGO in SR was relatively uniform from a macroscopic point of view. Figure 11b shows a picture of the cross-section of SR-Pd/SGO where the GO nanosheets were not clearly observed due to the non-stacking property of the SGO, thus endowing SGO with better compatibility with SR by silanization [21].

In order to study the thermal stability of silicone rubber and its composites, a thermogravimetric (TG) analysis was performed. As can be seen from Figure 12a, both Pd/SGO and Pd/C can improve the thermal stability of SR. In all three samples, the decomposition process took place at a temperature range between 100 and 400 °C. In this process, the Si-O bond was rearranged and broken, and then low molecular weight cyclic siloxane was generated and volatilized, resulting in the rapid degradation of SR [8]. The weight loss in the temperature range of 450 to 600 °C was caused by the decomposition of B_2_O_3_. When the filler was Pd/SGO, since the surface of SGO was wrapped with polysiloxane, the SR molecular chains were entangled on the surface of SGO. Thus, the movement of molecular chains was restricted, which improved the thermal stability of SR. When the filler was Pd/C, the rigidity of the polysiloxane molecular chains was increased through the action of activated carbon, which hindered the degradation of the chains [22]. In Figure 12b, all samples have three thermal decomposition stages, which were analyzed by DTG curves. The thermal weight loss in the first stage was caused by the silicon hydroxyl at the end of the silicone rubber molecular chain “back-biting” to the main chains. The second stage was caused by the random rupture of Si-O-Si chains in the SR [21]. In summary, the addition of both Pd/SGO and Pd/C can improve the thermal stability of SR.

Additionally, the self-healing properties of SR-Pd/SGO are also investigated and demonstrated in Figure 13. An intact SR-Pd/SGO sample was cut into pieces and put back into the mold. Then the SR-Pd/SGO was put into an oven at 70 °C for 4 h or room temperature for 12 h, and the self-healing behavior of the samples was observed. Almost no cracks were found on the surface of the sample. Since the SR-Pd/SGO polymer was viscoelastic at room temperature and the surface was a little sticky, self-healing was performed, when the composite was severely damaged. Due to the viscoelasticity, it can be processed and shaped in industry, which was beneficial for practical applications.

## 4. Conclusions

In this work, a new type of getter composed of Pd/SGO and SR was prepared and reported. The hydrogen elimination capacity of this new material is only dominated by the content of the vinyl group. The problem that Pd/C is difficult to disperse evenly in the polymer matrix is solved by the conversion from GO to SGO, where Pd nanoparticles are supported. Compared with commercial catalysts, the catalytic hydrogenation performance of Pd/SGO in SR has been greatly improved under various Pd concentrations. Moreover, the catalyst supported on SGO is expected to exert an improved performance in SR. The polymer elastomer we prepared performs excellent self-healing and processable properties, which are useful in scenarios where polymer integrity and high capacity are required. Due to the excellent performance of the getter, future work should be conducted in an effort to build a new getter with environmental adaptability and to resolve the structure–property relationship for the new material.

## Figures and Tables

**Figure 1 materials-15-04578-f001:**
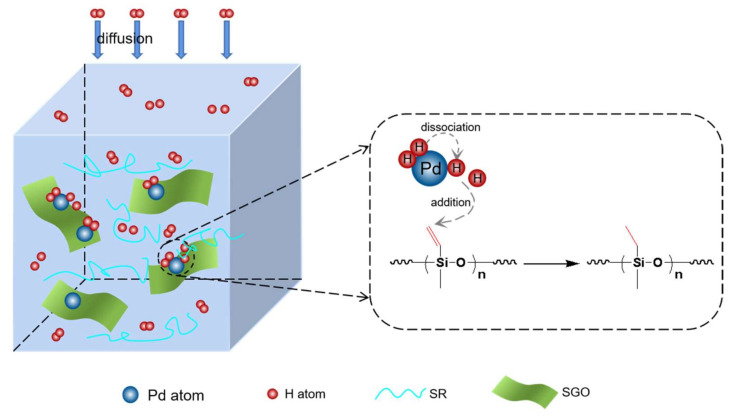
Hydrogen elimination mechanism of SR-Pd/SGO.

**Figure 2 materials-15-04578-f002:**
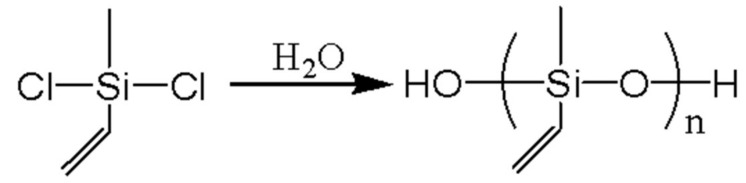
Preparation of PMVS by hydrolysis polycondensation.

**Figure 3 materials-15-04578-f003:**
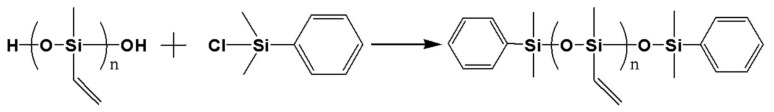
The reaction to quantification of molecular weight.

**Figure 4 materials-15-04578-f004:**
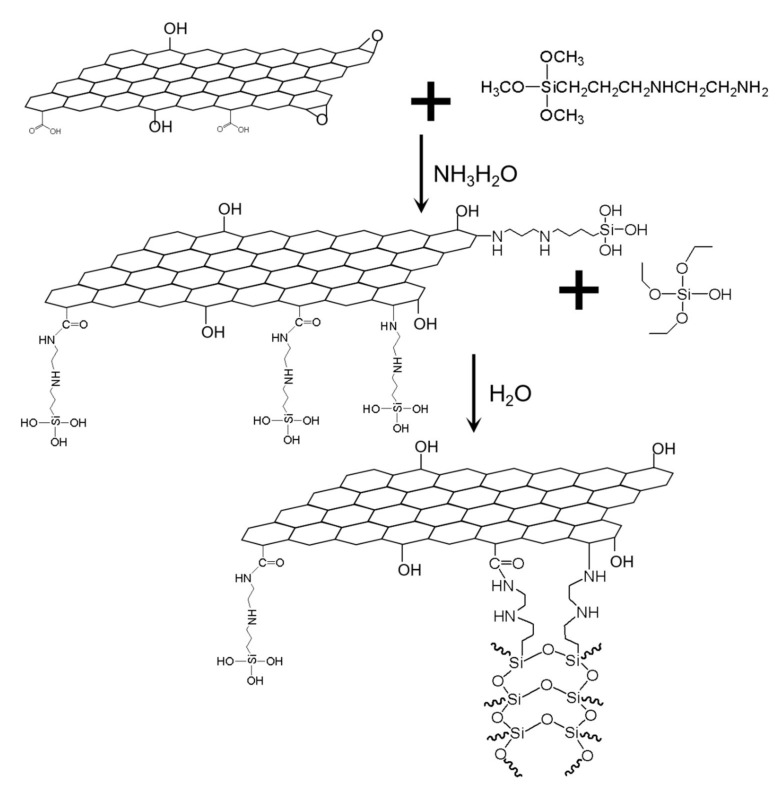
Formation mechanism of SGO.

**Figure 5 materials-15-04578-f005:**
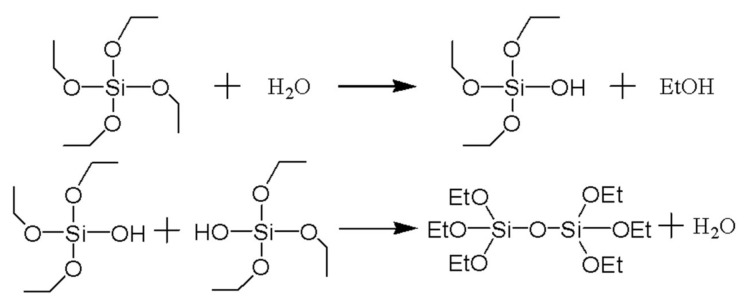
Hydrolytic condensation of TEOS.

**Figure 6 materials-15-04578-f006:**
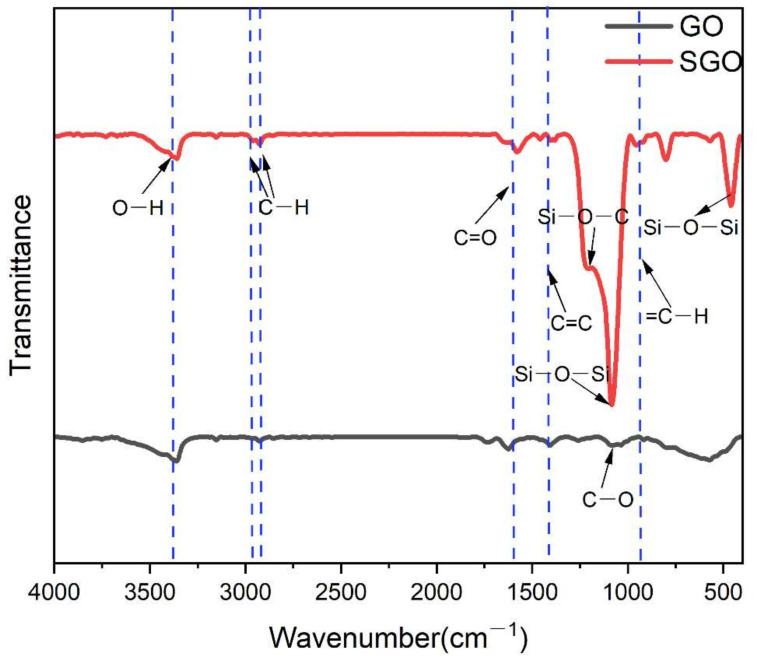
Infrared spectra of GO and SGO.

**Figure 7 materials-15-04578-f007:**
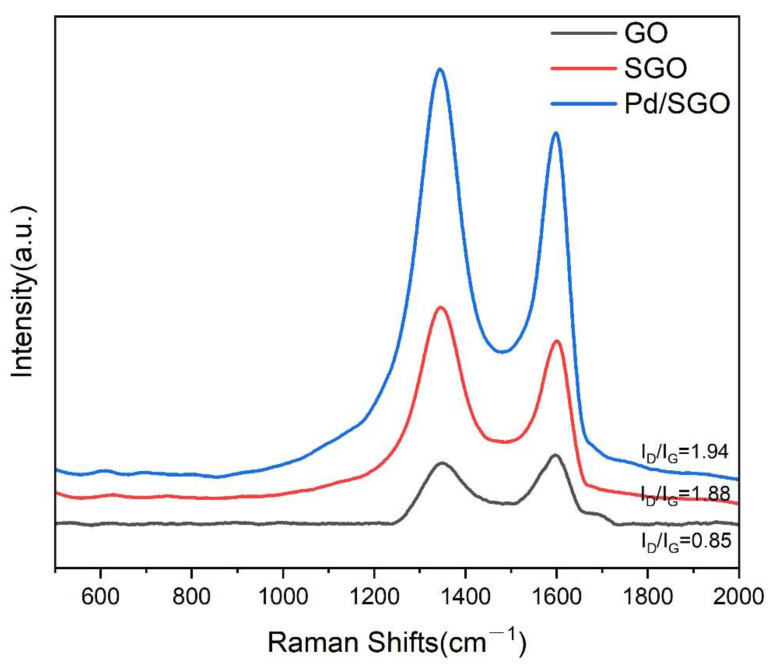
Raman spectra of GO, SGO, and Pd/SGO.

**Figure 8 materials-15-04578-f008:**
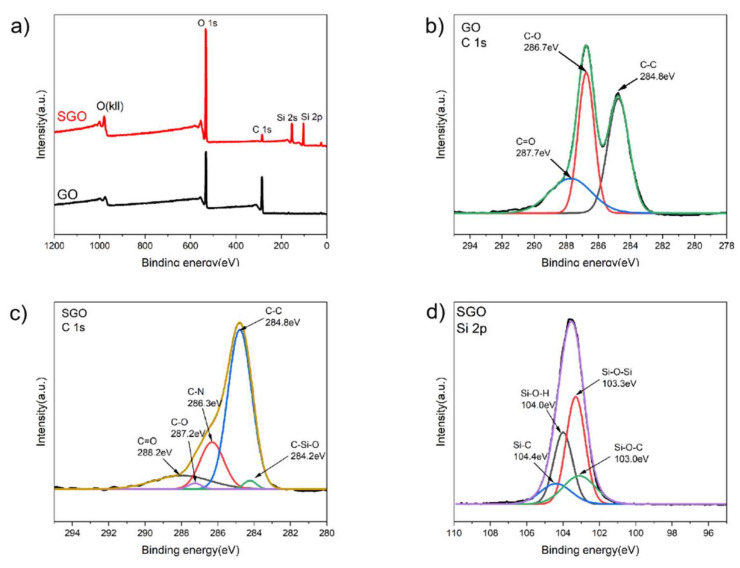
(**a**) XPS wide scan of GO and SGO. XPS C1s spectra of (**b**) GO and (**c**) SGO. (**d**) XPS Si2p spectra of SGO.

**Figure 9 materials-15-04578-f009:**
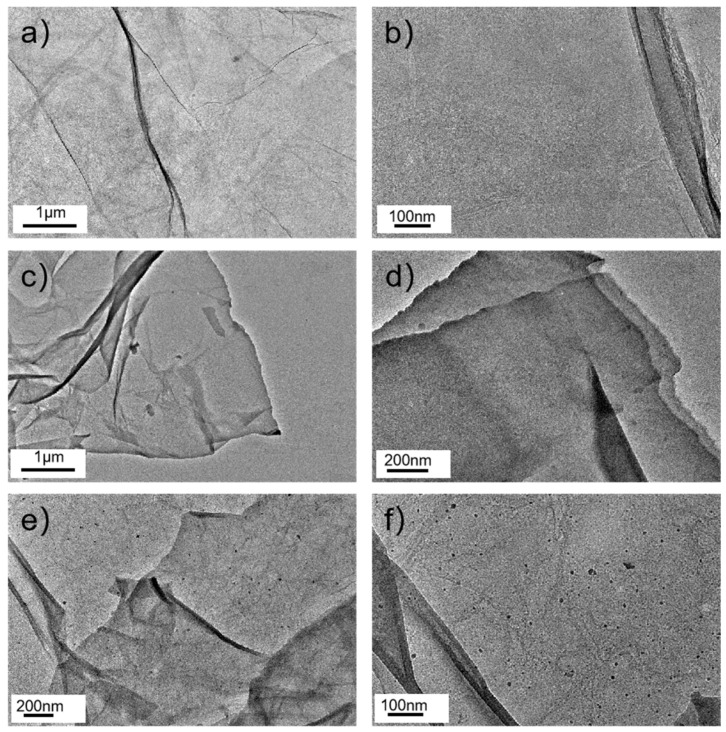
TEM images of (**a**,**b**) GO, (**c**,**d**) SGO, and (**e**,**f**) Pd/SGO.

**Figure 10 materials-15-04578-f010:**
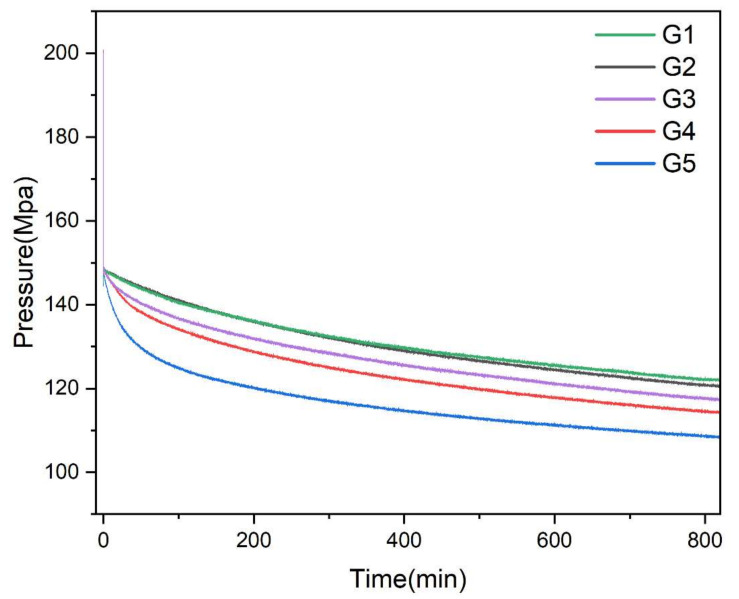
Hydrogen absorption test curves of G1, G2, G3, G4, and G5 samples in pure hydrogen.

**Figure 11 materials-15-04578-f011:**
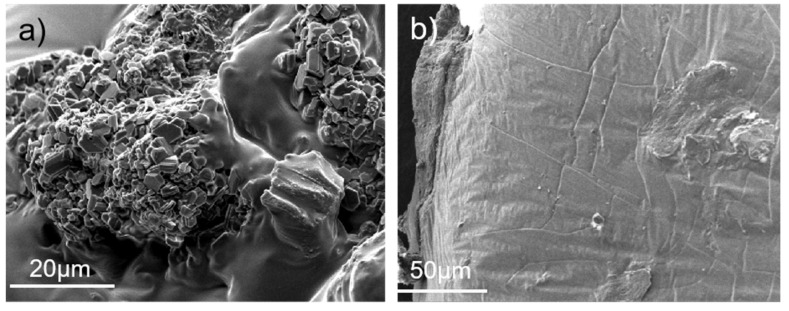
Cross-sectional SEM image of (**a**) SR-Pd/C and (**b**) SR-Pd/SGO.

**Figure 12 materials-15-04578-f012:**
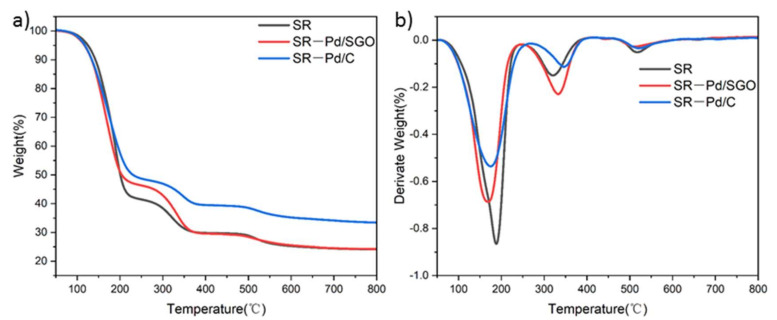
TG (**a**) and DTG (**b**) curves of pure SR, SR-Pd/SGO and SR-Pd/C composites.

**Figure 13 materials-15-04578-f013:**
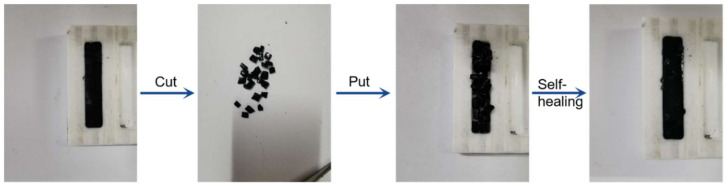
Self-healing process of SR-Pd/SGO.

**Table 1 materials-15-04578-t001:** The vinyl content of silicone oil.

Sample	Silicone Oil (mg)	Titrated Solution (mL)	Vinyl Content (mmol/g)	Theoretical Vinyl Content (mmol/g)
1	47	2.33	11.809	13.894
2	47	2.29	11.894	13.894
3	44	2.60	12.000	13.886
blank sample	0	7.88	-	-

**Table 2 materials-15-04578-t002:** Elemental content of GO and SGO measured by XPS.

Sample	Element Content (%)			
	C	O	Si	N
GO	72.8	27.1	-	-
SGO	12.9	56.7	30.2	0.3

**Table 3 materials-15-04578-t003:** Hydrogen absorption results of G1, G2, and G3 samples.

Sample	Hydrogen Absorption (/g SR)	Improved Capacity (Compared to G3)	Hydrogen Absorption (/mg Catalyst)	Improved Capacity (Compared to G3)
G1 (SR-3%Pd/SGO)	98.4 mL/g	−4.84%	32.8 mL/mg	58.45%
G2 (SR-Pd/C)	103.4 mL/g	0%	20.7 mL/mg	0%
G3 (SR-8%Pd/SGO)	118.4 mL/g	14.51%	29.6 mL/mg	43.00%
G4 (SR-10%Pd/SGO)	127.5 mL/g	23.31%	25.5 mL/mg	23.19%
G5 (SR-12%Pd/SGO)	150.6 mL/g	45.65%	25.1 mL/mg	21.26%

## Data Availability

Data is contained within the article.

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
