# Peer review of "Silanized Graphene Oxide-Supported Pd Nanoparticles and Silicone Rubber for Enhanced Hydrogen Elimination"

_materials, 2022, doi:10.3390/ma15134578_

Round 1
Reviewer 1 Report
This paper describes generation of new hybrid material for chemical absorption of hydrogen (new getter). The material is based on cross-linked polymethylvinylsiloxane. Hydrogen is absorbed by the reaction with vinyl groups of the polimer catalyzed by palladium Pd(0). The novelty is the specially prepared support of the catalyst - silanized graphen oxide which is also cross-linker of the polymer. The hybrid material obtained is self-healed. This paper may be published in Materials, however some descriptions are not clear and should be improved. The authors should take in consideration reviewer remarks as follows:
- The catalyst of the hydrogen addition to vinyl is Pd(0), while SGO is support, thus hydrogen absorption in Table 2 should be related to palladium not to Pd-SGO. The same should be done in the text in lines 246-247. The improvement in the performance of G3 in H adsorbed/g catalyst is 13.6% not 127.3%. The improvement is not so large as claimed by the authors.
- Since the concentration of Pd on the support play significant role, these studies should include investigations of more samples having a variable Pd concentration. The short list of samples is the weakness of this studies.
- Lines 101-1-2. The solution was stirred not the vessel was stirred.
- Lines 108-111. The description of the preparation of G1 is here described, not the preparation of Pd/SGO and SR composite.
- Line 127. The vinyl content should be shown in the paper. The content of vinyl shown in Table S1 is not clear. Show the content in mmol of Vi per 1g of the material (SR or hybrids).
- Line 138 and Equation 1. Phenyl groups are at both chain termini, thus either M2 should be multiplied by 2 or hydrogens of two phenyl groups should be taken into account which should be stated in the sentence in line 138.
- Figures 11 (a) and 11 (b) are not comparable as the first one is the image of surface while the other is the image of cross-section.
Author Response
Reviewer 1: This paper describes generation of new hybrid material for chemical absorption of hydrogen (new getter). The material is based on cross-linked polymethylvinylsiloxane. Hydrogen is absorbed by the reaction with vinyl groups of the polimer catalyzed by palladium Pd(0). The novelty is the specially prepared support of the catalyst - silanized graphen oxide which is also cross-linker of the polymer. The hybrid material obtained is self-healed. This paper may be published in Materials, however some descriptions are not clear and should be improved. The authors should take in consideration reviewer remarks as follows:
Reply: Thanks for your helpful comments and suggestions!
1. The catalyst of the hydrogen addition to vinyl is Pd(0), while SGO is support, thus hydrogen absorption in Table 2 should be related to palladium not to Pd-SGO. The same should be done in the text in lines 246-247. The improvement in the performance of G3 in H adsorbed/g catalyst is 13.6% not 127.3%. The improvement is not so large as claimed by the authors.
Reply: Thanks for your suggestion! As you said, the performance of hydrogen absorption (/g catalyst) should be calculated with Pd, not Pd-SGO. The performance of hydrogen absorption has been revised and added some other concentration of Pd.
2. Since the concentration of Pd on the support play significant role, these studies should include investigations of more samples having a variable Pd concentration. The short list of samples is the weakness of this studies.
Reply: Thanks for your suggestion! A whole set of data was tested and more hydrogen uptake data has been added with the different concentrations of Pd, as shown in Table 3 and Figure 10. Compared with G3, the hydrogen absorption (/g SR) performance of all samples was obviously improved, except G1 sample, which was caused by the low Pd concentration of G1. Even though the Pd content of G4 is the same as that of G3, the hydrogen absorption (/g SR) capacity was increased by 23.31%. Compared with G3, the hydrogen absorption (/mg catalyst) performance of all the samples was obviously improved. However, with the increase of Pd concentration, the improved capacity (compared to G3) was gradually reduced, which was caused by the catalytic efficiency of a single Pd atom decreasing with the increase of Pd concentration.
Table 3. Hydrogen absorption results of G1, G2, and G3 samples
Figure 10. Hydrogen absorption test curves of G1, G2, G3, G4, and G5 samples in pure hydrogen
3. Lines 101-1-2. The solution was stirred not the vessel was stirred.
Reply: Thanks for your suggestion! It is a mistake in my writing, which has been revised in the text. The sentence has been changed to ‘the solution was stirred at 70°C for 15 min’.
4. Lines 108-111. The description of the preparation of G1 is here described, not the preparation of Pd/SGO and SR composite.
Reply: Thanks for your suggestion! The description has been revised in the text, which was changed to the preparation of the getter, due to the preparation of multiple samples.
5. Line 127. The vinyl content should be shown in the paper. The content of vinyl shown in Table S1 is not clear. Show the content in mmol of Vi per 1g of the material (SR or hybrids).
Reply: Thanks for your suggestion! As you said, the content of vinyl is not clear, which has been revised in the text. We have shown the content in mmol of vinyl per 1g of the silicone oil. There is little difference between the actual content of vinyl and the theoretical content, as shown in Table 1.
Table 1. The vinyl content of silicone oil.
6. Line 138 and Equation 1. Phenyl groups are at both chain termini, thus either M2 should be multiplied by 2 or hydrogens of two phenyl groups should be taken into account which should be stated in the sentence in line 138.
Reply: Thanks for your suggestion! The hydrogen atoms of the phenyl group is an error due to my description. The sentence has been changed to ‘n1 is the number of hydrogen atoms in the two phenyl groups at the end of each molecular chain of the product’ in the text.
7. Figures 11 (a) and 11 (b) are not comparable as the first one is the image of surface while the other is the image of cross-section.
Reply: Thanks for your suggestion! It is also a descriptive error. In fact, Figures 11 (a) and 11 (b) are both the cross-section of the SR composite. We have revised the relevant description in the text. Figure 11 is the cross-sectional SEM image of (a) SR-Pd/C and (b) SR-Pd/SGO.
Reviewer 2 Report
The paper Silanized Graphene Oxide Supported Pd Nanoparticles and Silicone Rubber for Enhanced Hydrogen Elimination prepared by Yu Wang et al, presents interesting results that can be published after some improvements:
1. Please add at least 10-15 references from 2020-2021 in order to highlight better the importance and the novelty of your work in the field.
2. Figure 2 - seems that after reaction HCl is obtained. It must be an error and the supervisor did not check the scheme properly. Even if it is a nice joke, please re-draw the figure accordingly! Never ever you will obtain HCl from this kind of reaction. And from here could be 2 conclusions: a) it was an error, or b)...
3. Figure 6. Where the authors stated that there is C=C, again, seems to be a nice joke. Double bonds from where? There are the fingerprint regions of the compound(s), and most probable can be attributed to a different kind of C-H bending or deformation.
4. Could you define what is the difference between Hydrolytic condensation and hydrolysis and condensation? Thanks!
5. Figure 6 - again C=C from where?
Author Response
Reviewer 2: The paper Silanized Graphene Oxide Supported Pd Nanoparticles and Silicone Rubber for Enhanced Hydrogen Elimination prepared by Yu Wang et al, presents interesting results that can be published after some improvements:
Reply: Thanks for your positive comments!
1.Please add at least 10-15 references from 2020-2021 in order to highlight better the importance and the novelty of your work in the field.
Reply: Thanks for your suggestion! Lots of latest references have been added to the text, which makes my work more novel and convincing.
2. Figure 2 - seems that after reaction HCl is obtained. It must be an error and the supervisor did not check the scheme properly. Even if it is a nice joke, please re-draw the figure accordingly! Never ever you will obtain HCl from this kind of reaction. And from here could be 2 conclusions: a) it was an error, or b)...
Reply: Thanks for your suggestion! Maybe my understanding of the reaction is not very accurate, but the key part of the reaction is the hydrolysis condensation of chlorosilane rather than the formation of HCl. Figure 2 has been re-drawn in the text.
Figure 2. Preparation of PMVS by hydrolysis polycondensation
3. Figure 6. Where the authors stated that there is C=C, again, seems to be a nice joke. Double bonds from where? There are the fingerprint regions of the compound(s), and most probable can be attributed to a different kind of C-H bending or deformation.
Reply: Thanks for your question! Graphene skeleton is composed of many six-membered rings, each of which shows an absorption peak of C=C in the infrared spectrum. Since the catalyst was supported on the SGO, C=C is from the graphene skeleton. The absorption peak at 1575cm-1 was probable the skeleton vibration of C=C in the graphene.
4. Could you define what is the difference between Hydrolytic condensation and hydrolysis and condensation? Thanks!
Reply: Thanks for your question! In my opinion, hydrolytic condensation is a reaction in which the monomer is condensed while being hydrolyzed. When a chlorine atom is hydrolyzed to hydroxyl, it can condense with another hydroxyl, continuing step by step. However, hydrolysis and condensation are two separate steps. When all monomers are hydrolyzed, condensation is carried out.
5. Figure 6 - again C=C from where?
Reply: Thanks for your question! As mentioned above, C=C comes from graphene skeleton.
Round 2
Reviewer 1 Report
The authors improved the paper according to the reviewer remarks
Author Response
please find the revised manuscript in attachment
